

# Evaluation of the effectiveness of insecticide trunk injections for control of *Latoia lepida* (Cramer) in the sweet olive tree *Osmanthus fragrans*

Jun Huang[1,*], Juan Zhang[1,*], Yan Li[2], Jun Li[3] and Xiao-Hua Shi[1]

[1] Flower Research and Development Centre, Zhejiang Academy of Agricultural Sciences, Hangzhou, Zhejiang Province, China

[2] The Key Laboratory of Chemistry for Natural Products of the Guizhou Province, Chinese Academy of Sciences, Guizhou, Guiyang Province, China

[3] Guangdong Key Laboratory of IPM in Agriculture and Public Laboratory of Wild Animal Conservation and Utilization, Guangdong Institute of Applied Biological Resources, Guangzhou, Guangdong Province, China

[*] These authors contributed equally to this work.

Corresponding author
Jun Li, junlee100@163.com

## ABSTRACT

The screening of suitable insecticides is a key factor in successfully applying trunk injection technology to ornamental plants. In this study, six chemical pesticides were selected and injected into the trunks of *Osmanthus fragrans* to control the nettle caterpillar, *Latoia lepida* (Lepidoptera: Limacodidae), using a no-pressure injection system. The absorption rate of the insecticides, the leaf loss due to insect damage, and the mortality and frass amount of *L. lepida* larvae were evaluated after 77 and 429 days. The results showed that 4% imidacloprid + carbosulfan and 21% abamectin + imidacloprid + omethoate had the fastest conductivity and were completely absorbed into the trunks within 14 days; however, the efficiencies of these insecticides in controlling *L. lepida* were extremely low. Additionally, the treatment 10% emamectin benzoate + clothianidin and 2.5% emamectin benzoate was almost completely absorbed within 30 days and exhibited a longer duration of insecticide efficiency (>80% mortality) in the upper and lower leaves of the canopy. Treatment with these insecticides also resulted in significantly lower leaf loss and frass amounts. We conclude that emamectin benzoate and emamectin benzoate + clothianidin have a rapid uptake into *O. fragrans*, and are effective as insecticides over long durations. Hence, they may be a suitable control option for *L. lepida* in *O. fragrans* plants.

## INTRODUCTION

The sweet olive, *Osmanthus fragrans* (Thunb.) Lour., is a popular garden evergreen shrub or small tree that belongs to the family of Oleaceae. It has both ornamental and practical uses in landscaping and as incense (*Liu & Xiang, 2003*; *Lee, Lin & Yang, 2007*) and is widely planted in the Huaihe River basin and southern areas of China (*Wang et al., 2006*). The nettle caterpillar or blue-striped nettle grub, *Latoia lepida* (Cramer; Lepidoptera: Limacodidae), is distributed throughout Southeast Asia (*Azharul Islam et al., 2009*), especially in China,

Japan, India, Sri Lanka, Indonesia and Vietnam (*Hirashima, 1989*). *L. lepida* larvae mainly feed on *O. fragrans* leaves, resulting in restricted growth and dieback of leaves and twigs (*Wakamura et al., 2007*). Thus, this pest reduces the ornamental and practical values of the plants (*Ju et al., 2007*). In addition, exposure to the stinging spines on the dorsal surface of *L. lepida* can cause skin problems in humans, such as redness, swelling and pain, as well as other clinical manifestations such as fever, joint pain, and even death in allergic populations (*Qin, Li & Han, 1998*). Therefore, combating *L. lepida* infestations is both economically valuable and significant for protecting human health.

Currently, spraying chemicals on tree crowns is the main control method for *L. lepida* in China. However, chemical spraying can release pesticides into the air and water, and affect non-target animals, causing adverse consequences such as the deaths of large numbers of natural enemies, livestock poisoning, and environmental pollution (*Wakamura et al., 2007*). Chemical spraying is most commonly used in the green areas of cities or in the suburbs. In contrast, trunk injection technology is a more environmentally friendly method of applying pesticides, because it is highly efficient for liquid drugs, can be used with a broad spectrum of insecticides, and is relatively pollution free, safe, simple to apply, and is less affected by weather (*Navarro, 1992*; *Montecchio, 2013*). Trunk injection technology involves the injection of pesticides directly into tree trunks, which then transport the liquids through their conductive tissues to the site of action (*Mendel, 1998*; *Harrell, 2006*; *Mota-Sanchez et al., 2009*; *Doccola et al., 2011*); thus, trunk injection can play an important role in disease or insect pest control (*Mota-Sanchez et al., 2009*; *Takai et al., 2001*; *James, Tisserat & Todd, 2006*; *Darrieutort & Lecomte, 2007*). For example, using trunk injections of emamectin benzoate, ash trees with heavy infestations of *Agrilus planipennis* exhibited less canopy decline over a four-year period compared to non-treated control trees (*Flower et al., 2015*) and also resulted in a nearly 99% mortality of *A. planipennis* feeding on the treated tissues (*Smitley, Doccola & Cox, 2010*; *McCullough et al., 2011*; *Herms et al., 2009*).

Certainly, having a variety of insecticide options is a key factor in the successful application of trunk injection technology. *Byrne et al. (2012)* found that the uptake of 10% dinotefuran was more rapid than the uptake of 5% imidacloprid in California avocado groves. Both chemicals showed good control of the avocado thrips *Scirtothrips perseae*, and no residues were detected within the fruits. In contrast, although 10% acephate showed a rapid uptake and provided good control of thrips in bioassays, acephate residues and its insecticidal metabolite methamidophos were detected in fruits for up to four weeks after the injection. However, the uptake of 5% avermectin was slow, and it was ineffective against avocado thrips (*Byrne et al., 2012*). Another study found that trunk injections of imidacloprid, thiamethoxam and clothianidin in fully grown king mandarin trees to control the citrus greening disease vector *Diaphorina citri* resulted in approximately 50% mortality of the psyllids within 45 days. In general, imidacloprid had a better control effect than other insecticides tested (*Ichinose et al., 2010*). Therefore, evaluations of pest control using trunk injections of different chemicals provide a quick and effective assessment of the optimal trunk injection agent. However, little has been reported on the success of insecticide treatments using trunk injection techniques to control *L. lepida* on *O. fragrans* trees.
In this study, we selected six chemical pesticides to be injected, without pressure, into the trunks of *O. fragrans* to control *L. lepida*. First, the absorption rates of the insecticides were estimated on different observation days within a month after the trunk-injection application. In addition, the mortality of *L. lepida* larvae and tree leaf loss were evaluated in bioassays to determine the duration of efficacy at 77 days (approximately the period between two successive generations of *L. lepida* in a year in China) (*Ju et al., 2007*) and 429 days after treatment. Finally, we also investigated the amount of frass deposited by *L. lepida* larvae at these two time points. Our goal was to assess which type of insecticide performed best with regard to the uptake rate, efficacy against the target pest, and effective duration.

## MATERIALS AND METHODS

### Plants and insects

This study was conducted in a garden located in the Zhejiang Academy of Agricultural Sciences (30°18′75″N; 120°28′60″E), Hangzhou, China. The sweet olive trees used in this study, *O. fragrans* var. thunbergii, were 10–15 years old and planted in a total of three rows spaced approximately 3 m apart. The trees had well-structured crowns and a uniform growth trend. We randomly selected 21 individual trees and measured their heights, canopy widths, and diameter at chest height, resulting in means (±sem) of 4.53 ± 0.20 m, 2.39 ± 0.10 m, and 0.12 ± 0.01 m, respectively. These trees were managed with common watering and fertilization techniques; however, they were not subjected to chemical pesticides. Fifth instar larvae of *L. lepida* with similar weights were collected from sweet olive trees planted in the Hangzhou Blue Ocean Ecology Park (30°08′71″N; 120°31′49″E) and used for the bioassay. None of the study species are protected in China; therefore, no specific permits were required for collections or field activities.

### Insecticides

The insecticides used in this study included 95% imidacloprid and 70% emamectin benzoate (Guangdong Dafeng Plant Protection Technology Co., Ltd.), 95% abamectin (Hebei Weiyuan Group Co., Ltd.), 95% clothianidin emulsifiable concentrate (Nanjing Lebang Chemical Products Co., Ltd.), 98% omethoate (Lianyungang Dongjin Chemical Co., Ltd.), and 92% carbosulfan (Jiangsu Xingnong Co., Ltd.). These insecticides were diluted and formulated (or mixed) following the six trunk injection chemicals described in Table 1.

### Insecticide application by trunk injection

On 28 April 2014, 21 brown plastic bottles (Guangdong Institute of Applied Biological Resources supplied, designed by Dr. Li Jun) 6 cm high (from the bottom to the bottle neck) and 4 cm in diameter were prepared in the laboratory (Fig. 1). Each bottle was supplied with 30 mL of insecticide for trunk injection ($n = 3$ for each treatment). Three bottles were filled with distilled water (no insecticides) as controls. A hole approximately 30 mm in depth and 4 mm in diameter was drilled downward in the main trunk of each tree at a 45° angle approximately 30 cm above the ground using a rechargeable drill (Model TSR/1080-LI; Bosch Power Tools Co., Ltd., Shanghai, China). The bottle tip was cut open using a razor

**Table 1** Active ingredients and their formulation for the trunk injection.

| Trunk injection chemicals (abbreviation or code) | Formulation or composition | Active ingredient percentage (%) |
|---|---|---|
| EB + CL | Emamectin benzoate + clothianidin | 10 |
| A + I + O | Abamectin + imidacloprid + omethoate | 21 |
| EB | Emamectin benzoate | 2.5 |
| I | Imidacloprid | 4 |
| I + Ca | Imidacloprid + carbosulfan | 4 |
| EB + A | Emamectin benzoate + abamectin | 2.5 |

and inserted into the hole to completely inject the insecticides into the trunk. The screw threads on the tip of the bottle provide a good seal between the bottle and the edges of the drilled hole to prevent chemical leakage. Finally, an approximately 1 mm diameter air hole was made by puncturing the bottom of the bottle with an insect needle (approximately 0.75 mm in diameter and 40 mm in length) to promote the uptake of the insecticides. The quantity of residual agent in the bottles was visually observed and recorded at 9, 14, 23, and 30 days after application to test whether the absorption rates of the trunk-injected insecticides varied. During the assays, the temperature was $23.4 \pm 0.71$ °C and the atmospheric humidity was $67.8 \pm 2.10\%$; there were 5 rainy days (showers).

## Laboratory bioassay

Treated tree branches were sampled at 77 and 429 days after insecticide application and were brought to a laboratory to test the efficacy of insecticides on the targeted *L. lepida* larvae. Two branches from the bottom and top of the canopy were randomly collected from each tree in any compass direction. Each branch was 25–30 cm in length and had approximately 16 leaves. Debris and insects were removed from the branches and leaves before the test. A similar leaf-residue method (*Busvine, 1980*) and a custom setup were used for the larval bioassay (Fig. 2). The specific operational steps were as follows: (1) Each branch was placed vertically in a glass bottle (6 cm in diameter, 9 cm in height) filled with distilled water and sealed at the bottle neck with polystyrene foam; (2) the glass bottles were placed in the center of a plastic funnel (upper diameter, 40 cm; lower diameter, 5 cm; and height, 20 cm); and (3) the funnel was placed on the mouth of another glass bottle with the funnel neck (ca. 4 cm in length) inserted into the glass bottle and secured. This setup served to collect the larval frass in the bottom bottle. The inner wall of the funnel was coated with Teflon cream (Fluon®) to avoid the escape of fallen larvae.

*L. lepida* larvae (after being starved for 24 h) were allowed to stabilize for 12 h of observation before the test began. Two larvae were placed on each branch using a brush and allowed to feed on leaves for 5 consecutive days, during which time their frass was collected. The mortality of the larvae feeding on treated leaves was recorded after 5 days. The efficacy of the insecticide was evaluated based on the recorded mortality. Supplementary evaluation information was also recorded, such as the number of leaves eaten or damaged by larvae, and larval frass was weighed using an electronic scale (model EX223; Ohaus Inc., Parsippany, NJ, USA).

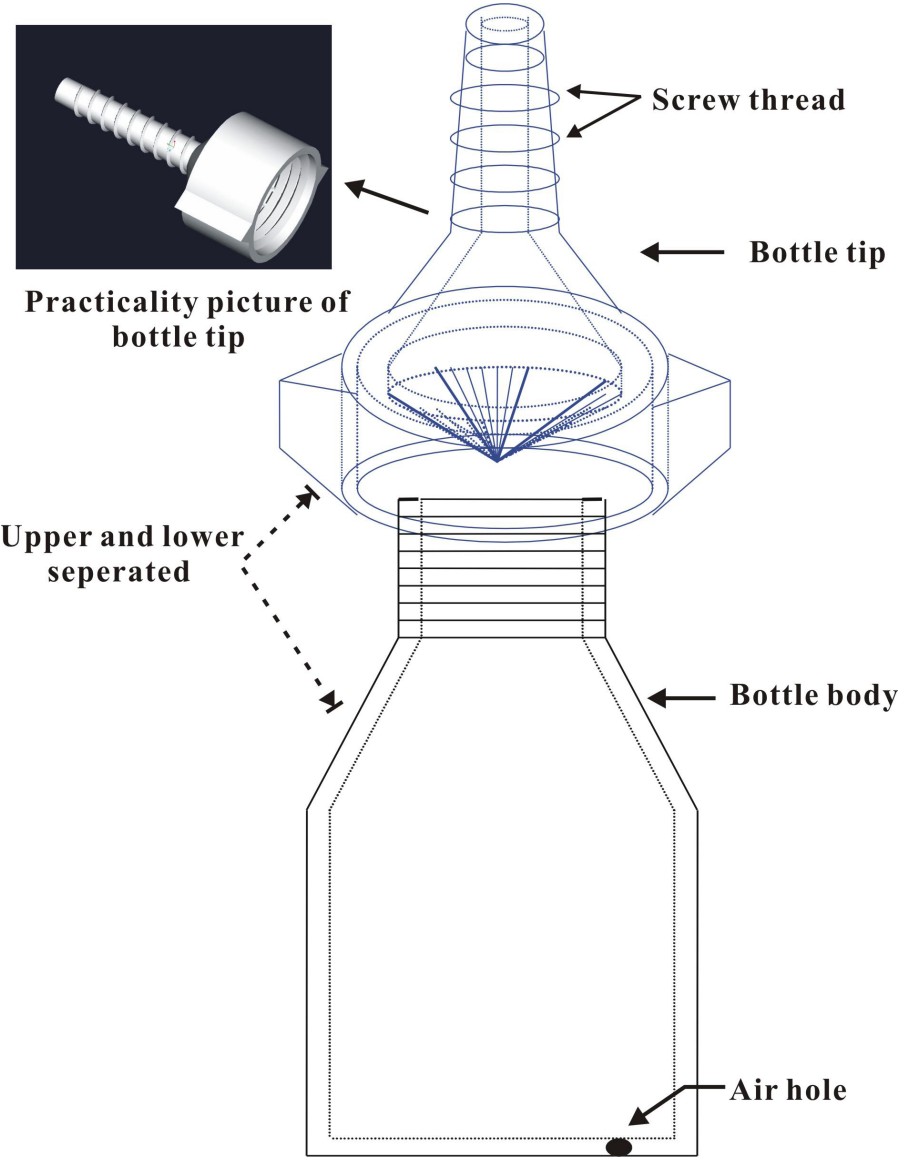

**Figure 1** **Structure diagram of the plastic bottle (injection device).** Two main parts of the bottle, i.e., bottle body (containing chemicals) and bottle tip (inserting the drilled hole).

## Statistical analyses

Shapiro–Wilks tests were applied to determine whether the data had a normal distribution and homogeneity of variance. When the data were normally distributed and exhibited similar variances, they were further analyzed using a repeated-measures ANOVA to compare the absorption rates between insecticides (between-subject) and between the examination days (within-subject). The mortality of larvae feeding on the upper and lower isolated branches following trunk injection with different insecticides at 77 days or 429 days was analyzed and compared using a two-way ANOVA and Duncan multiple range tests. The same methods were used to compare the differences in the percentages of damaged leaves and frass amounts between insecticides and leaf position (i.e., upper or lower

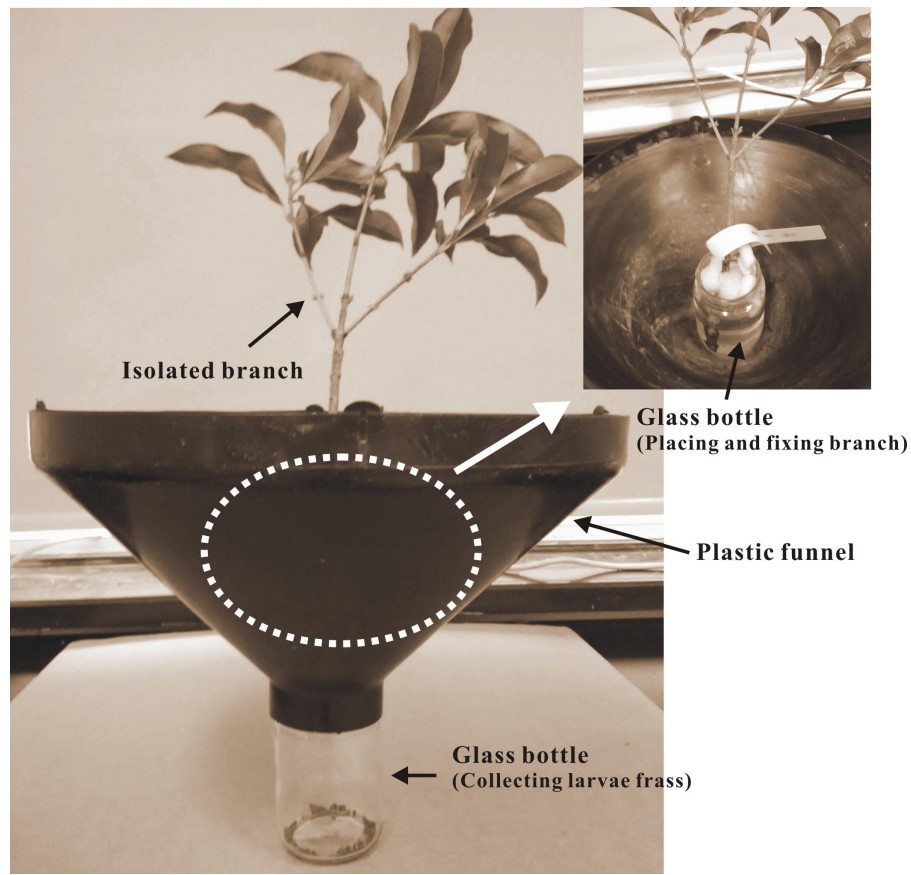

**Figure 2** A diagram of the setup for larval bioassay.

branches). When necessary, data were normalized by either square root or logarithmic transformations. All the statistical analyses were conducted using SPSS 14.0 (SPSS Inc., Chicago, IL, USA).

## RESULTS

### Absorption rates of the insecticides

The quantities of the six insecticides were reduced according to the number of days that had elapsed after the applications (Fig. 3). In particular, insecticide quantities decreased dramatically between the 14th and 30th days. The tests showed that the absorption rates of the insecticides differed significantly between insecticides ($F_{5,36} = 8.899$, $P < 0.001$) and observation times ($F_{3,36} = 14.568$, $P < 0.001$), but the interaction between these two factors was not significant ($F_{15,36} = 0.825$, $P = 0.686$). Within 30 days of the injection, 4 of the insecticides were completely absorbed into the trunks: A + I + O, EB, I + Ca and EB + A. Among these, I + Ca exhibited the fastest injection speed (it was completely absorbed within 14 days) followed by A + I + O and EB + A (23 days). However, only 77.5% of EB + CL and 56.7% of I were absorbed within 30 days. In addition, the absorption rate of EB + CL showed no significant differences in all the measured time points ($P > 0.05$). At the

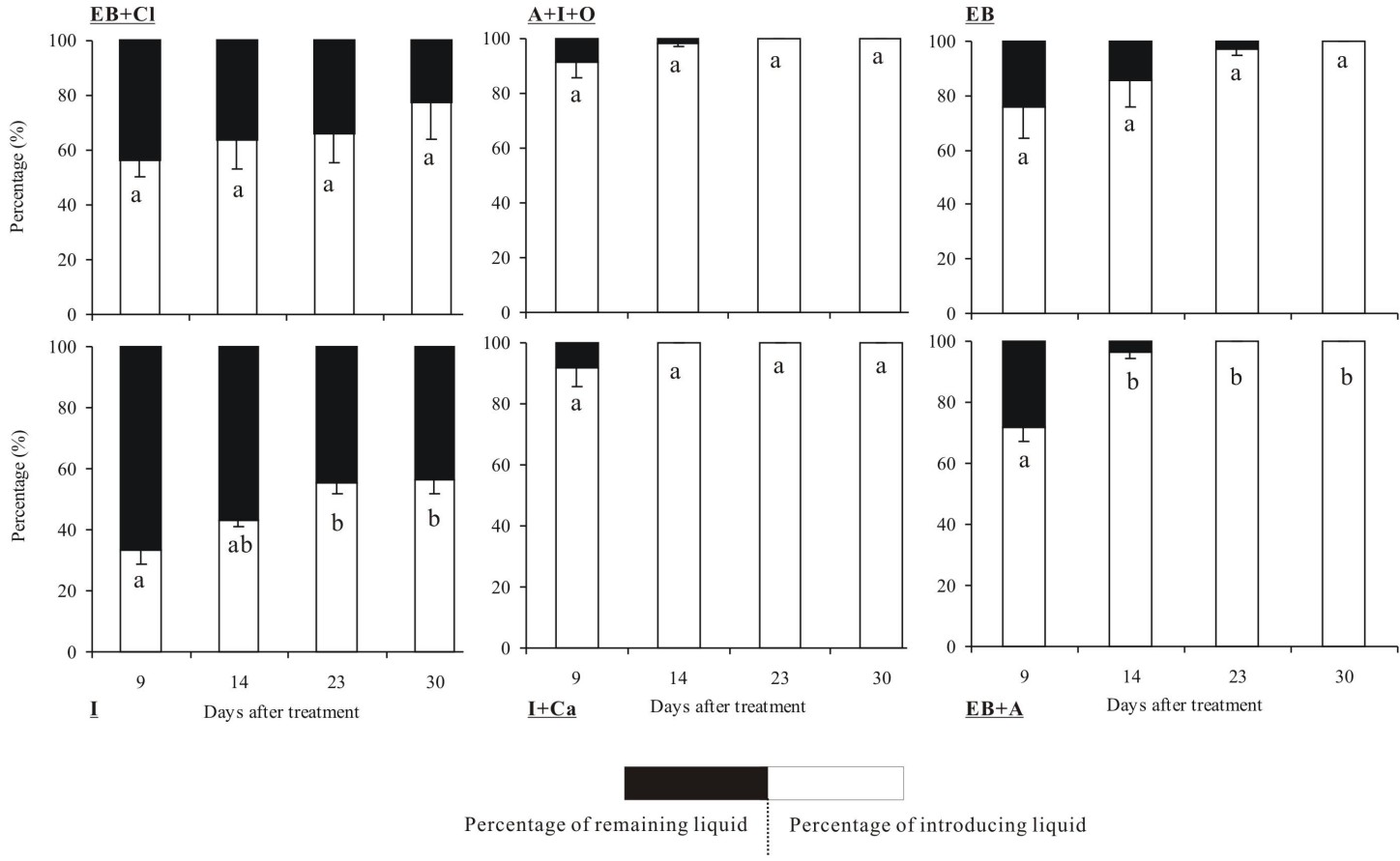

**Figure 3 Absorption rate of insecticides.** Mean (±SE) percentage (%) of the six types of insecticides (EB + Cl, A + I + O, EB, I, I + Ca, and EB + A) absorbed into the trunk of *Osmamthus fragrans* at 9, 14, 23, and 30 days after application of trunk injection. $N = 18$ trees. Insecticides (between-subjects) and observation time (within-subjects) effects were significant ($P < 0.001$); insecticides observation time interaction effect was not significant. Bars labeled with different lowercase letters are significantly different at $P = 0.05$ from each other in the same insecticide group based on Dunn's range test.

9-day point, the quantities of A + I + O and I + Ca absorbed were the largest (over 80% of the total).

## Larval mortality

Larval mortality after 77 days of treatment differed significantly between insecticides ($F_{6,28} = 23.721$, $P < 0.001$), but neither the leaf position nor the interaction between these two factors was significant ($F_{1,28} = 8.34$, $P = 0.007$; $F_{6,28} = 1.929$, $P = 0.111$; Fig. 4). Larval mortality from the EB + CL treatment was 100%, whereas mortality values from the A + I + O (0–33.3%), I + Ca (0), I (16.7–50%) and EB + A (33.3–66.7%) treatments were not significantly different from that of the control ($P > 0.05$); in fact, A + I + O and I + Ca caused no mortality. Larval mortality 429 days after treatment differed significantly between insecticides ($F_{6,28} = 14.878$, $P < 0.001$), but neither leaf position nor the interaction between these two factors was significant ($F_{1,28} = 0.031$, $P = 0.861$; $F_{6,28} = 0.454$, $P = 0.836$). Again, the mortality from the EB + CL treatment was 100%, while the data for A + I + O (16.7–33.3%) and I + Ca were not significantly different from the controls ($P > 0.05$),

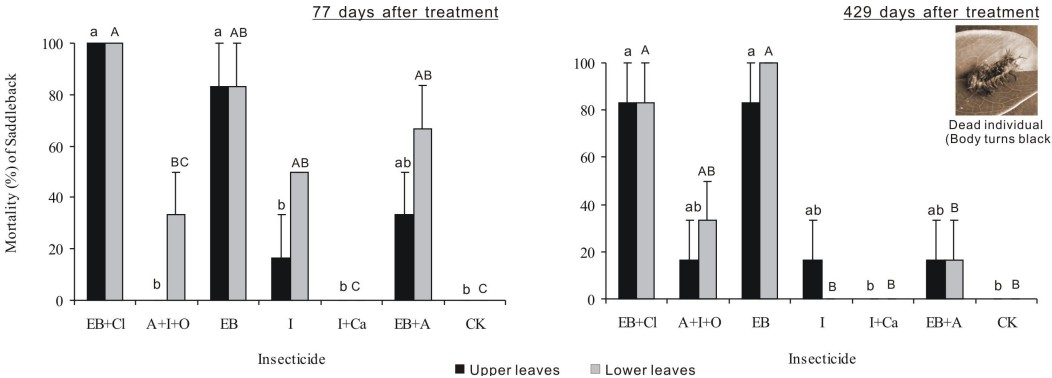

**Figure 4 Larval mortality.** Mean (±SE) mortality of *Latoia lepida* larvae on upper and lower leaves collected from *Osmamthus fragrans* trees treated with the insecticides of EB + CL, A + I + O, EB, I, I + Ca, and EB + A, by trunk injection after 77 and 429 days, respectively. $N = 42$ isolated branches in each observation time. Any observation time, insecticides effect was significant ($P < 0.001$); leaf position and insecticides × leaf position interaction effects were not significant. Bars labeled with different lowercase or uppercase letters are significantly different at $P = 0.05$ from each other in the same leaf layer group (upper or lower leaves) based on Dunn's range test.

especially for I + Ca (mortality = 0). These results indicate that although I + Ca had a good absorption rate after application, it had no insecticide efficacy on the larvae.

## Leaf loss

After 77 days of treatment, the percentages of damaged leaves were significantly different between insecticides, leaf position and the interaction between these two factors ($F_{6,28} = 19.439, P < 0.001; F_{1,28} = 43.969, P < 0.001; F_{6,28} = 8.921, P < 0.001$; Fig. 5). The percentage of upper leaves damaged, in total, was approximately 20% or less for EB and EB + CL and significantly less than that for the other treatments ($P < 0.05$). However, in comparison with the upper leaves, the data for lower damaged leaves for all the agents were significantly different from that of the controls ($P < 0.05$). The percentage of lower damaged leaves, in total, was less than 12% for EB + CL, A + I + O and EB (Fig. 3). Notably, the greatest contrast in damaged leaves from the upper (78.7%) and lower branches (8.2%) occurred with the A + I + O treatment. After 429 days, the percentage of damaged leaves was significantly different between insecticides ($F_{6,28} = 12.498, P < 0.001$), but neither the leaf position nor the interaction between these two factors was significant ($F_{1,28} = 3.603, P = 0.068; F_{6,28} = 0.76, P = 0.607$). The percentages of upper leaves damaged, in total, for EB + CL, EB, A + I + O and I were not significantly different from that of the control ($P > 0.05$); however, they were below the percentages from the other insecticides. The percentages of lower damaged leaves for EB + CL, EB and A + I + O were less than those from I and from the controls ($P < 0.05$).

## Larval frass

After 77 days of treatment, the frass amount differed significantly between insecticides ($F_{6,28} = 44.768, P < 0.001$), but neither the leaf position nor the interaction between these two factors was significant ($F_{1,28} = 1.837, P = 0.186; F_{6,28} = 0.424, P = 0.857$; Fig. 6). For all the treatments (except I + Ca), the frass amounts were smaller than controls ($P < 0.05$).

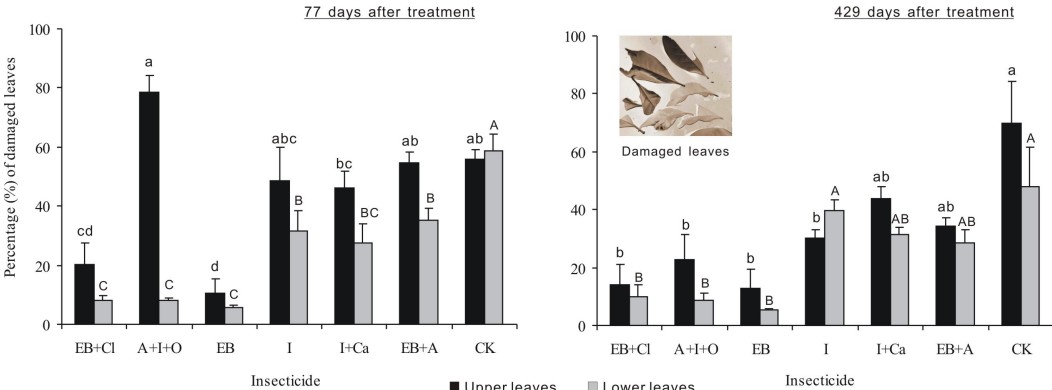

**Figure 5  Percentage of damaged leaves.** Mean (± SE) percentage of damaged leaves collected from upper and lower canopies after the application of six types of trunk injection at 77 and 429 days. $N = 42$ isolated branches in each observation time. 77 days after treatment, insecticides, leaf position, and insecticides leaf position interaction effects were significant ($P < 0.001$); 429 days after treatment, insecticides effect was significant ($P < 0.001$), and leaf position and insecticides × leaf position interaction effects were not significant. Bars labeled with different lowercase or uppercase letters are significantly different at $P = 0.05$ from each other in the same leaf layer group (upper or lower leaves) based on Dunn's range test.

For EB + CL and EB, the data were more obvious. After 429 days of treatment, the frass amount was significantly different between insecticides and leaf position ($F_{6,28} = 65.478$, $P < 0.001$; $F_{1,28} = 15.061$, $P < 0.001$), but the interaction between these two factors was not significant ($F_{6,28} = 4.935$, $P = 0.0015$). The frass amounts for EB and EB + CL were smaller than those found in the other treatments ($P < 0.05$). The frass amount from larvae on the upper leaves with I + Ca was significantly different from that of the controls ($P < 0.05$); however, for larvae on lower leaves, the frass amount was not significantly different from the controls ($P > 0.05$).

## DISCUSSION

The selection of appropriate trunk injection agents is key for the successful implementation of trunk injection technology (*Dedek et al., 1986*; *Takai, Suzuki & Kawazu, 2004*). For a no-pressure injection system (where the only pressure in the system is that of gravity), it is important for the liquid chemicals in external injection plastic bottles to move into the plants quickly; in other words, this is the first indication of how well the liquid chemicals have been absorbed after application. No-pressure injection systems such as the method used here may seem to be less advantageous because their lack of pressure can make the uptake slow; however, they are inexpensive and simple to use. Here, we found that four insecticides (i.e., A + I + O, EB, I + Ca and EB + A) were completely absorbed into the trunks within 30 days; additionally, more than 80% of A + I + O and I + Ca were absorbed into the trunks after only 9 days. However, for 4% imidacloprid, only 56.7% of the agent was absorbed within 30 days, and its insecticide efficacy on the mortality of *L. lepida* larvae was poor. In contrast, the conductivity and insecticide efficacy of imidacloprid on avocado groves (*Byrne et al., 2012*) and ash trees (*Mota-Sanchez et al., 2009*) were acceptable, although the authors did not mention whether the chemicals were completely absorbed into the plants after

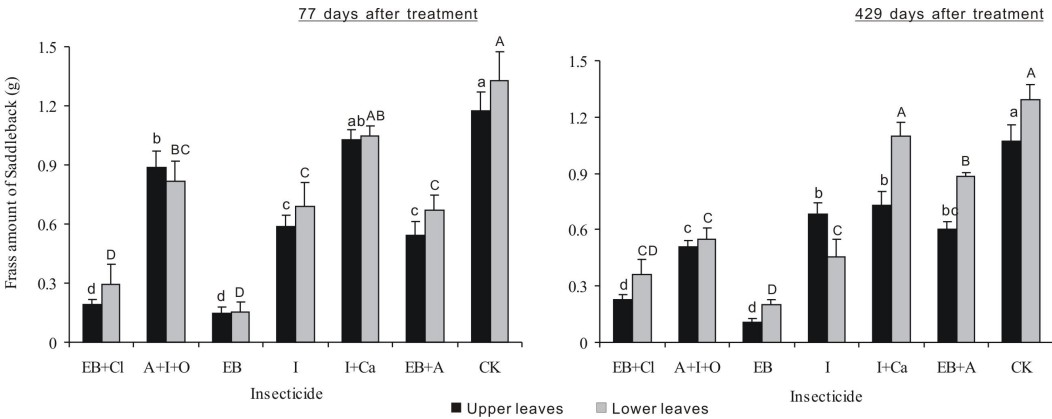

**Figure 6 Frass amounts of *L. Lepida* larvae.** Mean (± SE) frass amounts of *L. Lepida* larvae fed on upper and lower isolated leaves for 5 consecutive days after the application of six types of trunk injection at 77 and 429 days. $N = 42$ isolated branches in each observation time. 77 days after treatment, insecticides effect was significant ($P < 0.001$), and leaf position and insecticides × leaf position interaction effects were not significant. 429 days after treatment, insecticides and leaf position effects were significant ($P < 0.001$), and insecticides × leaf position interaction effect was not significant. Bars labeled with different lowercase or uppercase letters are significantly different at $P = 0.05$ from each other in the same leaf layer group (upper or lower leaves) based on Dunn's range test.

application. The reason for this result may be that chemical conductivity was affected by the injection time, procedure, tree size, growth, or even the type, concentration and formulation of the chemicals (*Harrell, 2006*; *McCullough et al., 2005*; *Cowles, Montgomery & Cheah, 2006*; *Tanis et al., 2012*).

It is worthy to note that larval mortality could be produced by using our trunk injection and bioassay method, because the insecticide residues had been detected in leaf samples collected from treated trees, e.g., 16.5–45.6 µg g$^{-1}$ emamectin benzoate occurred in upper leaf tissues after half a year (data now shown), although the sample number was not enough for statistics; while on the other hand, in the control group all larvae were able to maintain normal physiology state and carry out molt. We used the method of trunk injection to complete the conductance or movement of chemical compounds through the tree. The choice of application method was according to local climate conditions. In general, heavy rain occurs during April to early July in southeast of China. Other methods, such as leaf spraying and trunk painting, are more vulnerable to heavy rain, which washes off the insecticide applied to leaves or trunks (*Ichinose et al., 2010*), and the mortality of target pests was significantly fluctuating due to the different degree of rainfall following the application of leaf spraying (*Ichinose et al., 2010*). Meanwhile, the larval bioassay in our study was similar to a leaf-residue method (*Busvine, 1980*), which was contained on the host plant.

We found that EB alone or mixed with other agents (i.e., EB + A and EB + CL ) exhibited a better absorption rate and insecticide efficiency. Although only 77.5% of the total amount of EB + CL was absorbed into the injected trees, its insecticide efficiency, based on larval mortality, achieved a level as high as that of EB (>80%). We suggest that the mixture of EB and CL may have a synergistic effect. Interestingly, other chemicals such as A + I + O and I + Ca showed a better absorption rate but a lower insecticide efficiency. Specifically,

larval mortality was zero in the I + Ca treatment group, and the surviving larvae could enter the molt stage. However, previous studies have reported that imidacloprid insecticides effectively control many groups of insects, such as sap-feeders and beetles, following trunk injection (*Jeschke & Nauen, 2008*; *Mota-Sanchez et al., 2009*). The reason for the failures that we observed may be that (1) the chemical residue in the leaves was too low to be effective as an insecticide, or our trunk injection of A + I + O and I + Ca may not have provided sufficient volume for a duration of 77 days, and/or (2) although previous studies showed that chemical metabolites were toxic to target insects as well as the parent compound (*Nauen, Koob & Elbert, 1998*; *Mota-Sanchez et al., 2009*), it is possible that the effective components of the chemical may be negatively impacted by plant metabolic processes.

Previous studies have shown that the concentrations of trunk-injected chemicals among plant tissue types were different among plants as a whole, but that leaves showed much greater concentrations (*Mota-Sanchez et al., 2009*; *Takai, Suzuki & Kawazu, 2004*). For example, the imidacloprid concentrations in leaves increased steadily throughout the first growing season and were highest in leaf tissues, also were detected in leaves in the year following the injection (*Mota-Sanchez et al., 2009*). Therefore, for leaf-feeding insect pests, leaf loss was negatively correlated with the chemical concentration in leaves. EB (emamectin benzoate) acts as an antagonist for gamma-aminobutyric acid-gated chloride channels, causing a disruption of nerve impulses and rapid paralysis in a range of Lepidopteran species (*Kass et al., 1980*; *Ishaaya, Kontsedalov & Horowitz, 2002*). In addition, it has excellent control effects on nematodes (*Takai et al., 2000*; *Cheng et al., 2015*) and emerald ash borers (*Flower et al., 2015*) through either trunk or soil injections. Similarly, in our study, we found that the conductivities of both EB + CL and EB were acceptable, and they also had a longer duration of insecticide efficiency (429 days). However, another mixed agent, EB + A, showed insecticide efficacy only on the lower leaves and failed to persist over time. This result may have occurred because the different agent mixtures had different active ingredients in different concentrations. In the A + I + O treatment group, leaf loss from the lower canopy was less than that from the upper canopy, which indicates that higher concentrations of the agent were retained primarily in the lower leaves, or that degradation of the insecticide was higher in the upper canopy.

The amount of frass excreted by the insect pests can be used as the main indicator for estimating whether an insecticide is efficient (*Paguia, Pathak & Heinrichs, 1980*; *Yang et al., 2006*). In the present study, we found that the larval frass was affected to various degrees by all the treatments except for I + Ca; however, the frass amounts from the EB and EB + CL treatment groups were below those of the other treatments, which suggest that such chemical agents may have a stronger insecticidal effect on larvae. A previous study demonstrated that a decrease in food uptake was significantly correlated with decreased frass in insect pests (*Yang et al., 2006*). This result corroborates our previous investigation (J Huang, 2016, unpublished data), in which we found that the amount of frass was significantly positively correlated with the extent of leaf damage. Interestingly, insects can reduce the toxicity of chemical agents through an excretion mechanism (*Bues, Bouvier & Boudinhon, 2005*; *Liu et al., 2006*). Therefore, the detection and analysis of frass could be

an important method for further estimating the metabolic residues of injected chemical agents. However, the efficacy of insecticides based on the mortality of the targeted *L. lepida* is the most important prerequisite for choosing suitable trunk-injection insecticides.

## CONCLUSION

Overall, we conclude that emamectin benzoate (EB) and emamectin benzoate + clothianidin (EB + CL) trunk-injected insecticides were rapidly absorbed into *O. fragrans*, demonstrated significant insecticide efficacy against *L. lepida*, and remained effective over a longer duration than the other insecticides. However, the safety of these injection insecticides on the flowers of *O. fragrans* must be further studied in future research.

## ACKNOWLEDGEMENTS

Professor Wayne B. Hunter and other anonymous referees are thanked for useful comments to improve this manuscript. The authors are grateful for the assistance of Dr. Zhang Shao-yong (School of Forestry and Bio-technology, Zhejiang Agricultural and Forestry University) for his constructive comments regarding the insecticide trunk injections.

### Funding

This work was supported in part by the National Key Research and Development Program (2016YFC1201100, 2016YFC1201104), Key Programs of Agricultural Science and Technology of Xiaoshan (grant number, 2013203, 2015210) and the Funds for Environment Construction and Capacity Building of GDAS' Research Platform (2016GDASPT-0305, 2016GDASPT-0215). The funders had no role in study design, data collection and analysis, decision to publish, or preparation of the manuscript.

### Grant Disclosures

The following grant information was disclosed by the authors:
National Key research and Development Program: 2016YFC1201100, 2016YFC1201104.
Key Programs of Agricultural Science and Technology of Xiaoshan: 2013203, 2015210.
Environment Construction and Capacity Building of GDAS' Research Platform: 2016GDASPT-0305, 2016GDASPT-0215.

### Competing Interests

The authors declare there are no competing interests.

### Author Contributions

- Jun Huang conceived and designed the experiments, performed the experiments, analyzed the data, contributed reagents/materials/analysis tools, wrote the paper, prepared figures and/or tables.
- Juan Zhang performed the experiments, analyzed the data, wrote the paper, prepared figures and/or tables.

- Yan Li analyzed the data, contributed reagents/materials/analysis tools, reviewed drafts of the paper.
- Jun Li conceived and designed the experiments, contributed reagents/materials/analysis tools, prepared figures and/or tables, reviewed drafts of the paper.
- Xiao-Hua Shi performed the experiments, reviewed drafts of the paper.

### Data Availability

The raw data has been supplied as a Supplementary File.

### Supplemental Information

Supplemental information for this article can be found online at http://dx.doi.org/10.7717/peerj.2480#supplemental-information.

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
