# Peer review of "Evaluation of the effectiveness of insecticide trunk injections for control of Latoia lepida (Cramer) in the sweet olive tree Osmanthus fragrans"

_PeerJ, doi:10.7717/peerj.2480_

## Round 0.1 · original submission · Major Revisions

Both reviewers feel that the manuscript needs serious editing for English-please address those concerns prior to resubmission. In addition, the reviewers have put considerable efforts into helping point out major conclusions from the study that were lost in the discussion.

·

Basic reporting

Over all the authors have attempted a challenging study. However they need to correct several elements before the manuscript will be ready to be published. Some grammar issues need correction and results could be clarified for improving understanding. Comments, and markups on manuscript included. the Authors list insecticides then later in a table state Pharmaceutical drugs? Table 2?are being tested. Error bars on figure do not always match real values, and so some of these do not overlap where indicted there is not a difference, and some figures could have data stated in text, with P value. There was a correlation between lower mortality and more % of leaves eaten. More leaves eaten correlated with more frass produced. Thus more frass produced, the greater the leaf area consumed Fig 6. but the authors need to clarify conditions for results for the figure. The chemicals were scored, or evaluated on absorbance into the tree. The 'conductance' or movement of compounds through the tree in a systemic manner was then examined to result in insecticidal effects in the leaves of the upper and lower canopy. thus the main variable that needs to be explained thoroughly is whether or not mortality could be produced using this method. Otherwise how would this be better than topical sprays of these compounds ?

Experimental design

No statistic reported in the section for larvae mortality, which would be of most interest.
To evaluate movement the authors depend upon absorption into the tree trunk. Then insecticidal effects to show chemical has moved systemically through the tree. If they have access to a LCMS, or chemistry collaborator, they could have analyzed the leaves and shown chemical presence, but using insecticidal effects still shows product movement and presence. I think this is the point they are most focused on at the beginning of the text and then wander through and need to focus more on this aspect. see comments in manuscript.

Validity of the findings

Put P values in results section and in Figure captions, with brief explanation of what is important or what is the significant difference the authors want to point out to the reader.
reduce figures by one or two since feeding, frass, and leaf area are strongly correlated. They can state this in results, and then only discuss significant differences within each topic IF important.

Additional comments

Overall writing is average and fixable, but mislabeling of test classifications ie. insecticides versus pharma drugs, is a problem that must be corrected. Focus on the points you want to showcase (your best results), you do not have to discuss every tiny result. should try to eliminate one figure at least.

Reviewer 2 ·

Basic reporting

There is a problem with the level of English throughout the manuscript. I found it difficult to fully understand many of the sections within the manuscript and it required a lot of effort for me to decipher some of what the authors' were trying to portray.

EXAMPLES:
For leaf damage (Figure 3), you need to relabel from % bitten to % damaged. And in the text, you refer to it as a ratio (Lines 221, 225, 227, 230) - there is no description of a ratio that I can see, so I think you should replace the word ratio with the word percentage. It is the percentage of leaves damaged not the ratio of trees damaged.

Line 107/108: "(died for heavily Agrilus planipennis impacted)" does not read well
Line 127: "the leading quantity of chemicals was" - what do you mean by leading quantity of chemicals
Line 129: "nearly the span of the two generation larvae" - what does this mean?
Line 130: "duration of inter annual"
Line 290: correct the spelling of larval

Experimental design

The experimental section does not appear to be strong, and I think this is partly due to the poor quality of the English. The methods are not well described, and are way too brief. For a future submission, it is important that the authors describe the injection system in more detail.

In Table 1, it is not clear what the percentages of each chemical component are. Does 10% imidacloprid + clothianidin mean that each chemical formulation was diluted to 10 % of its original, or does 10% refer to the active ingredient percentage. This is a major area of confusion and needs to be clarified.
What type of bottle was used as the injection device (source)? How effective was the seal between the bottle and the drilled hole? Was there any leakage, potential for evaporation? One way around this would be to include a photo of the set up. Also, it seems that the insecticides used for the injections were not formulated for trunk injection. Diluting chemicals intended for foliar or soil application is not the norm for trunk injection and some justification for using this approach should be included in the discussion.
The larval bioassay method was difficult to understand, and needs to be clarified. Make a better distinction in your discussion of % leaves damaged (Figure 3) and total damage (Figure 4). Again, if there is an option to include a photo of the setup, then perhaps that might help. Better wording would be: "Branches were sampled at 77 d and 429 d after treatments, and insects were allowed to feed on these branches for 5 consecutive days during which the frass was collected." This is a straightforward method, but it is not clearly explained in the text.

Line 139/140: "tree spacing of approximately 30 cm" this means that the trees were planted 1 foot apart, yet they had canopy widths of over 2 meters. Can you clarify?

Validity of the findings

It appears that there were some positive effects of the trunk injections on larval mortality and leaf damage. I agree that the data presented shows EB and its mixtures to have performed better.

Line 272-274: "the effective components of the chemical were not compatible with the metabolites of plant because there are significant differences in the ability to metabolize exogenous compounds among different plant species." This sentence does not read well. I think the point you are trying to make is that the insecticides that did not perform well may have been impacted by plant metabolic processes. There is no evidence of this from your data, and ti would require measurement to confirm.

Additional comments

You have clearly put a lot of effort into this study. Unfortunately, you have not done your work sufficient justice n the final write-up. I found it difficult to read in many areas, and I found the methods were not described in sufficient detail for me to easily understand them. It is hard to interpret data when the description of the methods used are vague. I would recommend that when you revise your manuscript that you solicit the help of a colleague who has a better grasp of the English language. In that way, you will have a better chance to convey the results of your studies more effectively.

---

## Round 0.2 · Minor Revisions

You have done a great job in revising the manuscript according to reviewer comments. I have just a few more suggestions, mostly editorial, for you to consider. I have an edited word file that I will send to PeerJ for transmittal to you to facilitate the revision. I didn't find Table 2-please be sure to upload that with your files. There are no corrections on the other Table or Figures.

---

## Round 0.3 · accepted · Accept

Thank you for your diligence in improving the manuscript.